# Content of Phenolic Acids in the Grain of Selected Polish Triticale Cultivars and Its Products

**DOI:** 10.3390/molecules26030562

**Published:** 2021-01-21

**Authors:** Joanna Kaszuba, Ireneusz Kapusta, Zuzanna Posadzka

**Affiliations:** Department of Food Technology and Human Nutrition, Institute of Food Technology and Nutrition, College of Natural Sciences, University of Rzeszow, Zelwerowicza 4 St., 35-601 Rzeszow, Poland; ikapusta@ur.edu.pl (I.K.); zposadzka@ur.edu.pl (Z.P.)

**Keywords:** triticale, phenolic acids, ferulic acid, UPLC-PDA-MS/MS

## Abstract

The triticale grain has high nutritive value and good technological suitability. Triticale flour can be a valuable raw material for bread-making. The aim of this work was to determine the profile of phenolic acids in triticale grain of selected Polish cultivars and its products. Ultra-high-performance liquid chromatography (UPLC-PDA-MS/MS) was applied for separation and identification of these constituents. The grain of the examined triticale cultivars contained 13 phenolic acids, of which ferulic acid was determined in the largest amount and was constituted from 42–44% of the total content of phenolic acids in the grain. In addition, due to the large amounts of ferulic, di-ferulic, and sinapic acids, composition of the phenolic acids fraction in triticale grain of the tested cultivars varied in comparison with that of wheat and rye cultivars. In triticale flour, the number of phenolic acids was nearly 4 times lower than in the grain, as phenolic acids were removed along with bran, in which their proportion was almost 9 times higher than in the grain intended for grinding. The application of bran in the bread recipe resulted in a 3.5-fold increase in the fraction of phenolic acids compared to the bread produced from triticale flour without bran addition.

## 1. Introduction

Triticale (x*Triticosecale* Wittm. Ex A. Camus) is the first human-produced cereal obtained by crossing wheat (*Triticum*) and rye (*Secale*). Today, it is grown in many countries. Poland is the largest producer of triticale grain in the world. In 2018, its production in Poland reached 4.09 million tons, while global production amounted to around 12.80 million tons. Other important producers of triticale grain are Germany (1.93 million tons), France (1.38 million tons) and Belarus (1.01 million tons) [1].

Many cultivars of winter hexaploid triticale have high yield and are better environmentally adaptable than winter varieties of wheat. Moreover, many varieties of triticale are more drought tolerant and show better adaptation to acidic soil than wheat [2]. The triticale grain is used primarily as a feed grain [3], although, due to its high nutritive value [4] and good technological suitability, it is becoming increasingly popular for human consumption, for example, in flour production. Triticale flour can be a valuable raw material for bread-making [5,6]. Among the factors limiting its widespread use is high amylolytic activity and low gluten content as well as its unfavorable rheological characteristics of the triticale flour dough, which is viscous and not very elastic. In addition to its industrial use in milling and baking, triticale is also a raw material to produce malt and beer [7,8].

Triticale grain is a source of valuable bioactive components. It contains phenolic acids (PAs) [9,10] and resorcinol lipids [11,12], components with antioxidant and antibacterial properties which increase the pro-health properties of foods made of this cereal [13,14]. According to Irakli et al. [15] the total content of free and bound PAs is different among species. For example, oat (1832 μg/g) and maize (1158 μg/g), barley (497 μg/g), rye (360 μg/g) and triticale (470 μg/g), durum wheat (298 μg/g) and bread wheat (343 μg/g). In plant raw materials rich in polysaccharides and polyphenols, the interactions of these components often significantly affect their processing and digestion of products made from them [16].

The nutritional value of triticale grain creates prospects for food production and human consumption [4] also due to the content of health-promoting components, such as PAs. As has been proven in scientific research [9], the genetic factor shapes the phenolic acid profile [17], it is different compared to wheat and rye grain, and with the progress of breeding and the introduction of new varieties, they may also differ in grain phenolic profile. The aim of this work was to determine and compare the profile of PAs in triticale grain of selected Polish cultivars. The comparison of the profile of PAs and their content allowed to verify the diversity of the tested cultivars. The Borwo, Fredro, and Panteon cultivars were selected because the results of studies by other authors indicate their potential for use for food purposes [6,7]. The triticale products of tested cultivars, such as flour, bran, and bread, were compared in terms of the content of PAs. Ultra-high performance liquid chromatography (UPLC) with Mass Spectrometry was applied for separation and identification of these constituents.

## 2. Results and Discussion

### 2.1. Identification of Phenolic Acids in the Triticale Grain and Its Products

Table 1 shows the phenolic acids profile identified by LC-MS/MS for the free phenolic extracts of triticale and its products (see Appendix A). The chemical structure of identified phenolic acids in triticale was presented in Figure 1. The PAs were represented by peaks 1–13, including 3 hydroxybenzoic (peaks 1, 3–4) and 10 hydroxycinnamic (peaks 2, 5–13) derivatives. The ESI-MS signals at *m*/*z* 137 (peaks 1 and 3) and *m*/*z* 197 (peak 4) were identified as 4-hydroxybenzoic acid, 3-hydroxybenzoic acid, and syringic acid respectively, by comparing their retention time and MS spectra data with those of an authentic standard. The chromatogram of free phenolic acids (PAs) of triticale was presented in Figure 2. Upon fragmentation by MS/MS 4-hydroxy- and 3-hydroxybenzoic acids produced the ions at *m*/*z* 93 due the loss of CO_2_ from their respective precursor ions. This pattern of fragmentation was a characteristic feature of hydroxybenzoic acid derivatives as for other PAs. Syringic acid, on the other hand, first lost a water molecule, generating a major fragment ion at *m*/*z* 179 followed by a loss of carbon dioxide producing the other fragment at *m*/*z* 135. Gruz et al. [18] reported the same fragmentation pattern of these compounds in white wine.

Five different polyphenols in the category hydroxycinnamis acid derivatives namely caffeic *p*-coumaric, *o*-coumaric, ferulic, and sinapic were found. They were identified by comparing their retention times and characteristic MS spectra data with those of authentic standards. Accurate mass measurements and fragmentation pattern during MS/MS further confirmed their structural composition. The pseudomolecular ions of *p*-coumaric acid, *o*-coumaric acid (*m*/*z* 163) and ferulic acid (*m*/*z* 193) produced the major fragment ions at *m*/*z* 119 and 149, respectively, during MS/MS corresponding to the loss of carbon dioxide from the precursor ion. The other fragment generated during MS/MS of sinapic acid was at *m*/*z* 178 due to initial loss of a methyl group from the precursor ion. A similar fragmentation of the compounds was reported by Parejo et al. [19] in fennel extract.

The remaining hydroxycinnamic acid derivative, caffeic acid was identified by their accurate mass measurements and MS/MS spectral data. The tentative mass spectrum for caffeic acid showed the deprotonated molecule [M − H]^−^ ion at *m*/*z* 179 The major fragment ions produced by MS/MS analysis were *m*/*z* 161 and 135, corresponding to loss of water and carbon dioxide molecules, respectively, from the precursor ion

In addition to the ferulic acid. Several di-ferulic acid isomers were detected. Five compounds (**9**–**13**) exhibited a product ion at *m*/*z* 193 after expulsion of one ferulic acid moiety in the MS2 experiment (Figure 3). These findings were with agreement with previously investigations [20].

### 2.2. Content of Phenolic Acids in the Grain

Cereal grains are a rich source of bioactive compounds which, depending on their chemical structure, exhibit antioxidant properties [21]. Among them, PAs with free radical scavenging ability, comprise a large group. In the cereal grain they are concentrated in the outer layers [22].

The content of PAs in triticale grain of the cultivars which were examined (Table 2) was close to their content in the grain of the parental species (1.34–1.37 mg/g), which had been determined in previous studies on cereal grains [23]. Moreover, the findings of the authors cited revealed that the highest content of PAs is the factor that distinguishes rye and wheat grain from other cereals.

Thirteen PAs were detected in the grain of the triticale cultivars which were examined (Table 2), among which ferulic acid was predominant. According to previous studies [23,24], this acid along with *p*-coumaric acid occurs in cereal grain in the largest amounts, which agrees with our findings. The amount of ferulic acid in the examined triticale cultivars was within 42–44% of the total content of PAs in the grain. In addition to ferulic acid, the grains examined also contained significant amounts of sinapic acid and di-ferulic acid isomers (2 and 4). In the cultivar Borwo, sinapic acid constituted 14.6% of all the PAs, while the smallest percentage (10.9%) was found in the grain of the cultivar Panteon. In the triticale grain investigated, there were differences in the contents of di-ferulic (2) and no differences in the contents di-ferulic (4) isomers, which were respectively 13.0–15.0% and 9.2% to 10.3% of the number of PAs in the grain. The next was 4-OH-benzoic acid, found in similar amounts in the fraction of PAs in the triticale grain of the examined cultivars, fluctuating between 6.3% and 7.4% of the PAs content in the grain. In the triticale cultivars analyzed, significant differences were found in the content of *p*-coumaric acid. The grain of the cultivar Panteon had the smallest content of this acid, while its highest concentration was in the cultivar Borwo, twice exceeding the amount found in the cultivar Panteon. *o*-Coumaric acid was determined in trace amounts; its content in the grain of the cultivar Fredro was the highest; however, did not exceed 1% of the total content of PAs in the triticale grain of this cultivar.

The findings of other authors [25,26] confirmed quantitative differentiation of the phenolic acid fraction in grains of various cereal species. Also, Menga et al. [17] showed differences in the content of phenols in chemical extracts between species of cereals. In addition, wheat grain was found to have the highest amounts of ferulic acid, vanillic acid, coumaric acid, and sinapic acid. On the other hand, the studies of Andreasen et al. [27] on rye grain confirmed that the grain was the most abundant in ferulic, sinapic, and *p*-coumaric acid. In turn, according to Weidner et al. [28], who analyzed the content of PAs in triticale grain, this grain had the highest proportions of ferulic, coumaric, and caffeic acid, followed by sinapic acid in the total content of PAs. Thanks to studies on triticale grain [15], a profile of PAs was obtained, with the highest share of ferulic acid followed by sinapic and *p*-coumaric acid. Our studies demonstrated that triticale grain of the examined cultivars had slightly different composition of the phenolic acid fraction when compared to the results reported by the cited authors, referring to the content of PAs in the grains of parental species. In our experiment (Table 2), ferulic, di-ferulic, and sinapic acid were identified in the largest amounts. Such a difference in the composition of the phenolic acid fraction of triticale grains, in comparison with that of parental species, has been confirmed by previous studies [29,30,31], which indicate that the content of PAs in the grain is associated with the genotype. However, according to other authors [17,23], the location of the crop, and therefore environmental factors, also significantly affect the formation of PAs in the grain. This has been confirmed by our results against the results of other authors [15,28].

### 2.3. Content of Ferulic Acid in Triticale Products

Previous studies [25,26] showed that phenolic compounds are concentrated in the outermost layers of the grain, and during milling they are removed with the bran. Therefore, the flour derived from the milled triticale grain of the investigated cultivars was examined for the total content of PAs as well as ferulic acid content (Table 3).

Triticale flour which was produced from the tested cultivars by milling the grain with a moisture content of 12.5%, had similar values of PAs (Table 3). The triticale flour from the cultivars contained 334.3–359.4 μg/g DM of PAs. These values were significantly lower compared to the content of phenolic acid in the grain from which the flour was obtained (Table 2). They constituted from 26.2 to 29.4% of the content of PAs in the grain. Furthermore, in the triticale flour, the proportion of ferulic acids was ranging from 67.2 to 70.5% of the total content of PAs, being larger than the share of this acid in the total content of PAs in the examined triticale grain (Table 2).

The triticale flour and bran obtained from tested cultivars enables the production of good-quality bread, which is also confirmed by other authors [6,7,32]. In our study, the products such as flour, bran, and breads, obtained from the grain of tested triticale cultivars were analyzed of the contents of PAs (Table 3).

The content of PAs in the bran derived from milling grains of triticale cultivars was found to be 8.0 (Panteon) to 8.9 (Borwo) times higher than in the triticale flour obtained along with the bran due to grain-milling (Table 3). In the cultivars examined, the content of PAs in the bran was more than 2.3 (Fredro) and 2.4 (Borwo and Panteon) times higher than in the grain (Table 2). Both the flour and the bran had the highest proportion of ferulic acid: about 45.0% of the total content of PAs. The content of PAs in the bran of the triticale cultivars examined was close to the values reported by other authors [9], who found that triticale bran contained 2.71 mg/g DM of PAs; the content was much lower than in wheat bran (4.40 mg/g DM), but similar to rye bran (2.53 mg/g DM).

Rye and wheat belong to the cereals whose grain and products are most often consumed by people from Northern and Eastern Europe. The search for new ideas, for example, the development of new products based on cereals and their preserves, enables growth in the offer of cereal products with pro-health properties [33]. In view of the above, triticale flour may be the raw material for their production [6,32,33]. In our experiment, triticale bread with added bran, which could be included in the daily diet, was baked and examined for the content of PAs. Assuming that the content of PAs is an indicator of the bread’s potential pro-health properties, PAs were determined in the bread baked from a blend of triticale flour and bran (10% by baking blend weight) originated from examined cultivars. These bioactive compounds were also determined in the bread baked from triticale flour without bran, which served as a control regarding the bread enriched by adding bran (Table 3). PAs in the control triticale bread (without bran) obtained from the cultivars Fredro and Panteon were at a similar level: on average 9.25 mg/100 g of bread, and 10.98 mg/100 g of Borwo bread. The content of ferulic and total PAs in the triticale bread with bran was significantly higher compared to the control. The 3.4 (Panteon) and 3.5-fold (Borwo and Fredro) increase in the proportion of PAs found in the bread made of a blend of triticale flour and bran confirms that the bread may be enriched with bioactive components, for example, PAs, by adding bran, thus increasing their intake with the daily diet.

## 3. Materials and Methods

### 3.1. Materials

The material investigated consisted of the grain of winter triticale (x*Triticosecale* Wittm. Ex A. Camus) cultivars Borwo, Fredro, and Panteon, obtained from Plant Breeding Strzelce Ltd. Co. (Strzelce, Poland)—The Plant Breeding and Acclimatization Institute Group, Division in Borowo (Poland). Triticale grain and grain-milling products were stored in a cereal store (15 °C). The content of PAs was also analyzed in flours, triticale bran, and the crumbs of triticale bread.

### 3.2. Methods

#### 3.2.1. Process of Grain-Milling

The process of milling triticale grain with a moisture content of 12.5% was carried out in a Quadrumat Junior mill (Brabender, Germany) and the size of the mesh of the conical sifter was d = 280 µm according to AACC (American Association of Cereal Chemists) Method No. 26-50.01 [34]. After milling, flour yield (%) was determined, i.e., the amount of flour obtained due to grinding 100 g of grain. The dry matter (DM) content in the grain, flour, and bran was determined in accordance with the AOAC (Association of Official Analytical Chemists) method No 925.10 [35]. Grain-milling and DM content determinations were made in triplicate.

#### 3.2.2. Determination of Water Absorption of Flour

The water absorption of flour at the maximum dough consistency of 350 FU was determined by a Brabender farinograph-E (Brabender, Germany) according to ICC-Standard No. 115/1 [36], in three repetitions.

#### 3.2.3. Bread-Baking Procedure

The baking of triticale bread from flour of the cultivars which were examined was performed by a single-phase method in triplicate. Dough was made of flour (600 g, moisture content of 14.0%), yeasts (3.0% by weight of flour), salt (1.5% by weight of flour) and water (to farinographic consistency 350 FU). In the triticale dough recipe, there was triticale bran in the amount of 10% by weight of the flour, i.e., 60 g of bran and 540 g of flour with a moisture content of 14.0%. The dough prepared according to this recipe was then fermented for 1 h in a fermentation chamber (at 30 °C and relative humidity of 85%) and pierced after 30 min. Afterwards, 250-g pieces of dough were formed, put into pans, and subjected to fermentation in a fermentation chamber allowing the dough to rise until reaching the optimal dough expansion. The process of baking was carried out at 230 °C for 30 min in a baking chamber of the electric modular Classic oven (Sveba Dahlen, Sweden). The loaves left to cool down to room temperature. The moisture content of the bread crumb was determined 24 h after baking, according to AACC Method No 44-15.02 [34].

#### 3.2.4. Extraction and Separation of Phenolic Acids

The triticale grain and bran were milled using a LabMill mill (Perten, Hägersten, Sweden) equipped with a homogenizing sieve, the mesh size of which was d = 0.5 mm. The bread crumb was freeze-dried for 48 h in an Alpha 1–2 LDplus laboratory lyophilizer (Martin Christ Gefriertrocknungsanlagen GmbH, Osterode am Harz, Germany) and then ground to particle granulation ≤0.5 mm in diameter.

In the first stage of extraction, the basic hydrolysis of a 2-g sample in a 4 M NaOH solution was carried out, maintaining the boiling temperature for 2 h. PAs were extracted from the hydrolysates by the method of Mpofu et al. [37] with a modification according to Żuchowski et al. [38]. The extraction was performed in triplicate.

The profiling of changes in phenolic acid concentration was done based on UPLC-PDA-MS/MS using a Waters ACQUITY UPLC liquid chromatograph (Waters Corporation, Milford, MA, USA), equipped with a photodiode array detector (PDA) combined with a mass spectrometer with a double quadrupole analyzer (Waters ACQUITY^®^ TQD (Tandem Quadrupole Detector), Micromass, Wilmslow, UK). The following parameters were used for TQD: capillary voltage 3.5 kV; con voltage 30 V in negative mode; the source was kept at 250 °C and desolvation temperature was 350 °C; con gas flow 100 L/h; and desolvation gas flow 800 L/h. Argon was used as collision gas at a flow rate of 0.3 mL/min.

Samples were separated at 50 °C on a Waters ACQUITY UPLC^®^ HSS C18 column (2.1 × 100 mm, 1.8 μm). The mobile phase consisted of eluent A (0.1% formic acid solution in water, *v*/*v*) and eluent B (0.1% formic acid solution in acetonitrile, *v*/*v*). The solvent gradient was programmed as follows: 5 min, 0% B; 0.5 min, 1% B; 2.5 min, 10% B; 10 min, 10–100% B; 1 min, 100% B; 1 min, 100–0% B. The flow rate was kept at 0.3 mL/min, the injection volume was 5 µL. All determinations were performed in triplicate.

#### 3.2.5. Identification of Phenolic Acids

Identification of PAs was based on the analysis of spectra of the phenolic acid standards (4-OH-benzoic, caffeic, 3-OH benzoic, syringic, *p*-coumaric, *o*-coumaric, ferulic, sinapic), obtained thanks to the chromatographic analysis. The specific spectra of the maximum UV absorbance, the mass-to-charge ratio (*m*/*z*) and the fragmentation spectra resulting from collisionally induced dissociation (CID) were compared. The di-ferulic acid isomers (1–5) were identified based on CID spectra.

#### 3.2.6. Quantitative Analysis of Phenolic Acids

For quantitative analysis, the Multiple Reaction Monitoring (MRM) experiment was used, in which the presence of fragment ions deriving from the selected precursor was monitored. Phenolic acid content was calculated based on the calibration curves of the peak area dependence on the concentration of substance injected on the column. The content of PAs in the grain, flour, and bran was converted into DM content and reported as mean values from three replicates. The content of PAs in the bread was expressed per 100 g of fresh bread and reported as the average of three replicates. Concentration of the PAs in samples was calculated on the base of standard curves by injection of solutions of known concentrations ranging from 0.05 to 5 mg/mL (R2 ≤ 0.9998) of phenolic acid as standards.

### 3.3. Statistical Analysis of Results

Statistical analysis of the results was done using the Statistica ver. 13 (TIBCO Software Inc., Palo Alto, CA, USA). All analyses were performed in triplicate and results were expressed as mean ± SD. The results obtained were subject to single-factor analysis of variance (ANOVA) and considering significant *p* < 0.05. The significance of differences between averages were verified by the Duncan test at the significance level *p* < 0.05.

## 4. Conclusions

Triticale grain is an important source of PAs. However, a significant decrease in the content of this group of bioactive compounds is observed in the flour obtained in the milling process. At the same time, the proportion of PAs in triticale bran is much larger than in the grain. Therefore, bran can be used as an additive to meals such as breakfast cornflakes and muesli, or may also be used as a valuable raw material in the production of bread with a higher bioactive value.

Considering the widespread consumption of bread and its availability, triticale bread enriched with bran seems to be a very good solution to supplement the diet with bioactive ingredients. For consumers accustomed to wheat and rye bread, it may be a new offer on the market of bakery products. In addition, in the regions with intensive production of triticale, such a possibility may favor the use of good-quality triticale grains for flour production, at the same time reducing its use as forage.

## Figures and Tables

**Figure 1 molecules-26-00562-f001:**
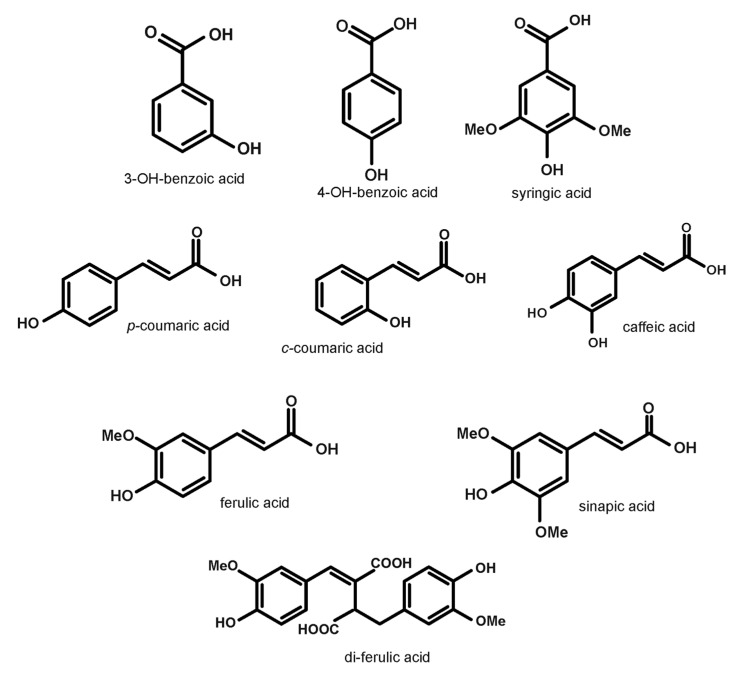
The chemical structure of identified phenolic acid in triticale.

**Figure 2 molecules-26-00562-f002:**
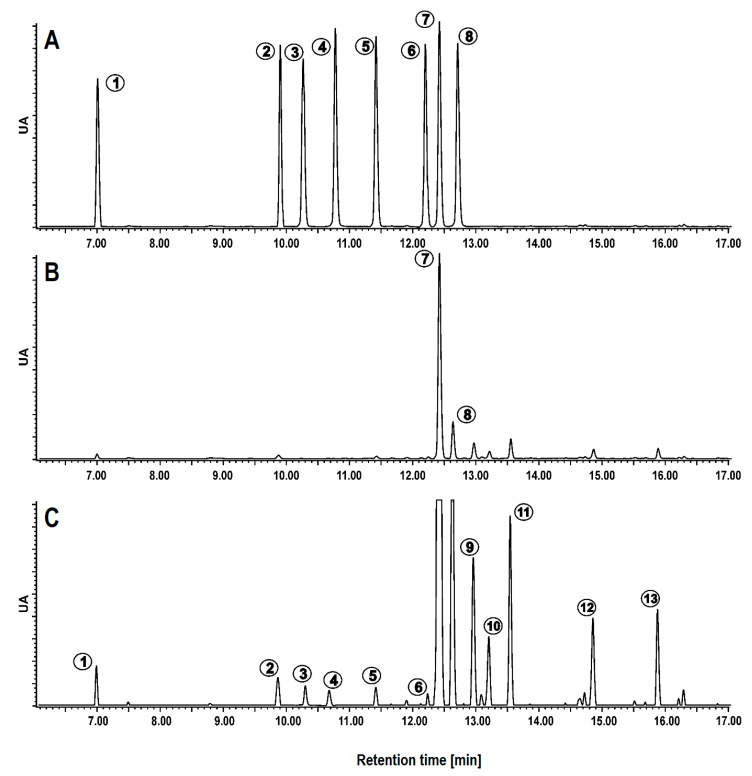
The chromatogram of free phenolic acids (PAs) of triticale. (**A**)—Mixture of eight phenolic acid standards, (**B**)—Chromatogram of triticale extract, (**C**)—Chromatogram of triticale enlarged 6 times to show minor compounds, (**1–13**)—Compounds’ numbers correspond to Table 1.

**Figure 3 molecules-26-00562-f003:**
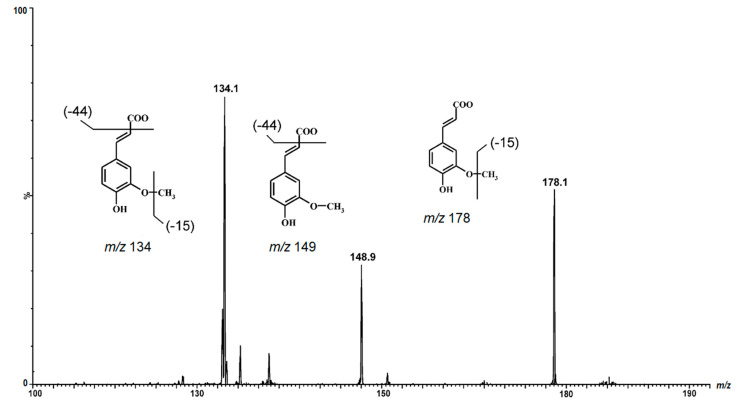
The ferulic acid fragmentation reaction in MRM transition.

**Table 1 molecules-26-00562-t001:** Separation parameters of phenolic acid standards by UPLC/MS method.

No.	Phenolic Acid	Retention Time	[M − H]^−^	Fragment Ion	Λ_max_	MRM Transitions	
						Quality Transition	Collision Energy
		[min]	[*m*/*z*]	[*m*/*z*]	[nm]	[*m*/*z*]	[eV]
1.	4-OH-benzoic	7.49	137	93	230	137→93	10
2.	caffeic	9.87	179	161, 135	320	179→163	20
3.	3-OH-benzoic	10.37	137	93	223	137→93	20
4.	syringic	10.73	197	179, 135	277	197→135	30
5.	*p*-coumaric	11.42	163	119	312	163→119	30
6.	*o*-coumaric	12.12	163	119	312	163→119	30
7.	ferulic	12.43	193	149, 134	322	193→134	30
8.	sinapic	12.63	223	179, 149	320	223→179	30
9.	di-ferulic (isomer 1)	13.20	385	297	323	385→297	30
10.	di-ferulic (isomer 2)	13.55	385	245	320	385→245	30
11.	di-ferulic (isomer 3)	14.63	385	319	322	385→193	30
12.	di-ferulic (isomer 4)	14.85	385	193	325	385→193	30
13.	di-ferulic (isomer 5)	15.68	385	293	322	385→193	30

**Table 2 molecules-26-00562-t002:** Content of phenolic acids in the grain of the examined triticale cultivars.

Examined Feature		Cultivar		
		Borwo	Fredro	Panteon
Content of phenolic acid [μg/g DM]	4-OH-benzoic acid	86.7 ± 6.4 ^a^	91.1 ± 5.1 ^a^	88.4 ± 5.7 ^a^
	caffeic acid	41.0 ± 3.4 ^a^	37.2 ± 1.2 ^a^	53.5 ± 3.5 ^b^
	3-OH-benzoic acid	10.9 ± 1.7 ^a^	14.4 ± 0.7 ^b^	14.5 ± 0.8 ^b^
	syringic acid	8.6 ± 1.2 ^b^	9.7 ± 1.3 ^b^	5.0 ± 0.6 ^a^
	*p*-coumaric acid	74.5 ± 6.1 ^c^	62.6 ± 3.9 ^b^	36.8 ± 0.8 ^a^
	*o*-coumaric acid	4.0 ± 1.3 ^b^	8.9 ± 1.2 ^c^	0.0 ± 0.0 ^a^
	ferulic acid	594.3 ± 32.5 ^b^	532.8 ± 20.1 ^a^	533.6 ± 28.2 ^a^
	sinapic acid	200.1 ± 17.5 ^b^	192.7 ± 16.4 ^b^	131.4 ± 5.4 ^a^
	di-ferulic acid (isomer 1)	25.9 ± 2.3 ^a^	22.0 ± 2.7 ^a^	24.2 ± 3.4 ^a^
	di-ferulic acid (isomer 2)	185.8 ± 10.9 ^b^	164.9 ± 7.2 ^a^	179.9 ± 3.3 ^ab^
	di-ferulic acid (isomer 3)	10.2 ± 0.5 ^a^	8.8 ± 0.7 ^a^	11.2 ± 3.7 ^a^
	di-ferulic acid (isomer 4)	125.5 ± 15.9 ^a^	127.3 ± 12.6 ^a^	124.0 ± 2.9 ^a^
	di-ferulic acid (isomer 5)	7.6 ± 3.9 ^b^	0.0 ± 0.0 ^a^	1.9 ± 0.8 ^b^
Total content of PAs [mg/g DM]		1.37 ± 0.09 ^b^	1.27 ± 0.07 ^a^	1.20 ± 0.04 ^a^

^a,b,c^—mean values followed by the different letters in the same line differ significantly at the significance level *p* < 0.05.

**Table 3 molecules-26-00562-t003:** The content of PAs in triticale products.

Examined Feature		Cultivar		
		Borwo	Fredro	Panteon
Content in flour [μg/g DM]	ferulic acid	253.2 ± 12.2 ^b^	231.8 ± 9.1 ^a^	237.6 ± 11.2 ^ab^
	total phenolic acids	359.4 ± 15.1 ^a^	334.3 ± 23.8 ^a^	353.4 ± 21.8 ^a^
Content in bran [mg/g DM]	ferulic acid	1.54 ± 0.02 ^b^	1.36 ± 0.05 ^b^	1.29 ± 0.05 ^a^
	total PAs	3.21 ± 0.11 ^b^	2.90 ± 0.13 ^a^	2.84 ± 0.11 ^a^
Content in bread [mg/100 g]	ferulic acid	5.41 ± 0.12 ^a^	5.21 ± 0.23 ^a^	5.09 ± 0.13 ^a^
	total phenolic acids	10.98 ± 0.16 ^b^	9.38 ± 0.90 ^a^	9.11 ± 1.01 ^a^
Content in bread with bran [mg/100 g]	ferulic acid	18.01 ± 1.12 ^b^	15.18 ± 1.31 ^a^	16.78 ± 1.52 ^ab^
	total phenolic acids	38.27 ± 2.02 ^b^	32.45 ± 2.14 ^a^	30.65 ± 2.46 ^a^

^a,b^—mean values followed by the different letters in the same line differ significantly at the significance level *p* < 0.05.

## Data Availability

The data presented in this study are available on request from the corresponding author.

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
