# Peer review of "Content of Phenolic Acids in the Grain of Selected Polish Triticale Cultivars and Its Products"

_molecules, 2021, doi:10.3390/molecules26030562_

Round 1

Reviewer 1 Report

The authors have analyzed phenolic contents in three triticale cultivars and their products (flour, bran, bread, and bread with bran). Although the results are interesting, but the results are too preliminary to be published in Molecules. In addition, the analysis of phenolics in triticale are already available, for instance (https://doi.org/10.1016/j.jff.2008.09.009). 

I think the result of this paper might be more suitable with other journals in MDPI such as FOODS, however needs more important results, for instance the effects of some selected compounds, cultivars on diabetes, or other diseases, or some specific and more valuable compounds.

Author Response

Dear Reviewer,

Thank you very much for your suggestion and comments.

The authors decided to submit their manuscript to Molecules due to the recruitment of articles for a special issue entitled “Bioactive compounds of fruits, vegetables and mushrooms”. The authors concluded that the results presented in this article fit the subject matter and provide valuable data on the influence of genotypes variability on the content of polyphenolic acids in triticale grains and its products.

The authors do not agree with the objection that the obtained results are only preliminary. Complete qualitative and quantitative analyzes of the profile of phenolic acids, which are the main group of polyphenols occurring in cereals, were carried out, using a specialized LC-MS method in the multiple reaction monitoring mode (MRM). The profile was investigated both in grains, but also the dynamics of changes in various products such as flour, bran and bread was taken into account, which determined the impact of the used technologies.

Poland is one of the largest producers of triticale in the world, mainly intended for feed for farm animals, however, there are more attempts to use triticale as a food raw material for humans. Therefore, we believe that the obtained results are valuable from the point of view of nutritional purpose.

We agree with the reviewer's postulate that research in this aspect requires greater involvement and determination of the activity of the obtained products in terms of the prevention of civilization diseases.

We are currently conducting more comprehensive research related to the bioavailability of polyphenolic compounds using a simulated digestive system and antioxidant activity. We are planning to publish the results in more relevant journals related to nutrition.

Reviewer 2 Report

Dear Authors the manuscript I have read, entitled "Content of phenolic acids in the grain of selected Polish triticale cultivars and its products". I think it should be published in the journal Agronomy of this MDPI Editorial, since the contribution in chemistry is not very relevant, but the nutritional and agronomic contribution is. The work is well written and well presented. But my decision is to change the journal so that it will not be published in Molecules.
Best Regards

Author Response

Dear Reviewer,

Thank you very much for you valuable comments.

The authors decided to submit their manuscript to Molecules due to the recruitment of articles for a special issue entitled “Bioactive compounds of fruits, vegetables and mushrooms”.  The authors believe that the content of the article fits the topic of SI and contains important content due to the analysis of polyphenolic compounds present in triticale, which show biological potential.

We have searched the Molecules database by the keywords and in the results there are nearly 700 articles on the profiling of phenolic acids in various plant matrices, especially fruits and herbs, so in our opinion presented manuscript will be a valuable supplementation to knowledge about the presence of polyphenols in cereals and cereal products.

Reviewer 3 Report

The paper is potentially interesting. However, I have several concerns. Please see the comments below.

-Abstract: the main part of the abstract reports just the description of the results, without any reference to the aims of the work and the methodologies used.

- The introduction should be improved by a more comprehensive literature review. In addition, authors must clearly identify the contributions of this paper in terms of the novelty and how these relate to previous studies.

- The aims of the study are not clearly stated. Why did you select these cultivars? What did you expect? A sort of hypothesis must be formulated.

- Materials and methods: A lot of important information is missing.  Sampling: How many samples were used?  How did you store the material? The exact number of analytical replicates must be specified in all material and methods section. Big parts of the LC-MS method description are missing, i.e. injection volume, gradient used, ionization parameters etc. Your data are in DRY WEIGHT basis. How did you calculate that?

-Results and discussion: A representative chromatogram for each cultivar should be reported.  There is not any discussion about the mass fragmentations obtained. Some sentences are unclear, and some parts of the discussion are poor and need to be redeveloped.

Please see the enclosed file for detailed comments

Author Response

Dear Reviewer,

Thank you for the constructive and informative comments of the reviewer.

-Abstract: the main part of the abstract reports just the description of the results, without any reference to the aims of the work and the methodologies used.

The abstract has been edited in accordance with the reviewer's comment and the aim of the work and the methodologies were added.

- The introduction should be improved by a more comprehensive literature review. In addition, authors must clearly identify the contributions of this paper in terms of the novelty and how these relate to previous studies.

Line 31: “Many cultivars of winter hexaploid triticale have high yield and are better environmentally-adaptable than winter varieties of wheat [2].”

Introduction: The reviewer's remark was included and supplemented in the text.

- The aims of the study are not clearly stated. Why did you select these cultivars? What did you expect? A sort of hypothesis must be formulated.

Line 44-45: The aims of study of the study have been clarified and the explanation was added.

- Materials and methods: A lot of important information is missing.  Sampling: How many samples were used?  How did you store the material? The exact number of analytical replicates must be specified in all material and methods section. Big parts of the LC-MS method description are missing, i.e. injection volume, gradient used, ionization parameters etc. Your data are in DRY WEIGHT basis. How did you calculate that?

According to the reviewer’s suggestion, the description of the raw materials and methods were supplemented in the text of manuscript.

-Results and discussion: A representative chromatogram for each cultivar should be reported.  There is not any discussion about the mass fragmentations obtained. Some sentences are unclear, and some parts of the discussion are poor and need to be redeveloped.

Please see the enclosed file for detailed comments

The reviewer's comments were taken into account. The subtitle “Identification of phenolic acids in the triticale grain and its products” was added. The chromatograms and a paragraph of mass fragmentation was added. The reviewer's comments were taken into account and the discussion paragraph was redeveloped.

Round 2

Reviewer 1 Report

The authors have tried to revised their paper. I do not reject in their method of using UPLC/MS to detect the presence of the 13 phenolic acids. However, why the quantification in the bread and bran was conducted only on ferulic acid, not the others? What does it mean "total phenolic acids"?  it is the total amount of the 13 phenolic acids you detected or all putative phenolic acids in the 3 triticale cultivars?

I know that in Poland, triticale is important crop. However, this study only quantified quantities of common phenolic acids in grains of the 3 cultivars and ferulic acid in bran and bread. Neither biological activities nor special food capacities of these cultivars involved in the phenolic acid profile are conducted. I think only in this step, it is difficult to be published in Molecules. I remind here that your results are too preliminary to be publishable. 

Author Response

Dear Reviewer,

thank you for your informative remarks.

Determination in bread and bran was carried out only on ferulic acid, because the content of this acid was its amount was close to or more than half of the phenolic acid content. The phrase "total phenolic acids" - this is the total amount of 13 phenolic acids detected in tested triticale cultivars.

Reviewer 2 Report

Dear authors: I understand and appreciate your justification that this chapter is special, but what is missing for example is a figure with the structures of table 1. However well known they may be, it should not be forgotten that this is a journal of chemistry. You can use chemsketch or chemdraw acd labs to draw the structures.

In addition, it would be elegant to mention an example of fragmentation of one of the mass compounds, which would give a chemical approach to the article.

Author Response

Dear Reviewer,

thank you for your comment on the manuscript. Figure 1 with the structures of Table 1 was added in accordance with them. According to the reviewer’s suggestion, an example fragmentation of one of the mass compounds (ferulic acid) was added (Figure 3).

Reviewer 3 Report

In this second version of the manuscript the authors added more details. However, I found other issues that need to be clarify before the aceptance of the manuscript.

Please see the enclosed file.

Author Response

Dear Reviewer,

thank you for the constructive and informative comments.. The reviewer's remark was included and supplemented in the text. According to reviewer's suggestion suplementary files have been added: Suplementary I: LC-MS/MS spectra of identified phenolic acids in triticale grain and its products. Suplementary II: UPLC chromatograms of triticale grain, flour.